# Trends and inequalities in full immunization coverage among two-year-olds in Sierra Leone, 2008–2019

Umaru Sesay[1,2], Augustus Osborne[3]*, Hassan Benya[4], Camilla Bangura[3],
Bright Opoku Ahinkorah[5,6]

1 Sierra Leone Field Epidemiology Training Program, National Public Health Agency, Western Area, Freetown, Sierra Leone, 2 Africa Field Epidemiology Network, Western Area, Freetown, Sierra Leone, 3 Department of Biological Sciences, School of Basic Sciences, Njala University, PMB, Freetown, Sierra Leone, 4 Ministry of Health, Western Area, Freetown, Sierra Leone, 5 REMS Consultancy Services, Takoradi, Sekondi-Takoradi, Ghana, 6 Faculty of Health and Medical Sciences, The University of Adelaide, Adelaide, Australia

* augustusosborne2@gmail.com

## Abstract

### Background

Immunization is a cornerstone of public health, preventing numerous vaccine-preventable diseases. However, disparities in immunization coverage persist globally, particularly in low-income countries like Sierra Leone. This study aims to examine the trends and socioeconomic and geographic inequalities in full immunization coverage among two-year-olds in Sierra Leone between 2008 and 2019.

### Methods

This study utilised data from the Sierra Leone Demographic Health Surveys conducted in 2008, 2013, and 2019. The Health Equity Assessment Toolkit software developed by the World Health Organisation was used to calculate various inequality measures, including Difference, Ratio, Population-Attributable Risk, and Population-Attributable Fraction. An assessment of inequality was conducted across six stratifiers: maternal age, maternal economic status, maternal education level, place of residence, child sex, and sub-national province.

### Results

Between 2008 and 2013, the coverage of full immunization coverage among two-year-olds in Sierra Leone increased from 41.1% in 2008 to 73.1% in 2013; however, the coverage decreased to 50.4% in 2019. The inequality associated with maternal age, sex of the child, and sub-national level increased between 2008 and 2019. However, the results were not statistically significant for maternal age and sex of the child.

**Data availability statement:** Third party data was obtained for this study from The DHS Program. Data may be requested from The DHS Program after creating an account and submitting a concept note. More access information can be found on The DHS Program website (https://dhsprogram.com/data/Access-Instructions.cfm). The authors confirm that interested researchers would be able to access these data in the same manner as the authors. The authors also confirm that they had no special access privileges that others would not have.

**Funding:** The author(s) received no specific funding for this work.

Similarly, inequalities existed with economic status, education, and place of residence steadily decreased over the same period, but the results were also not statistically significant.

## Conclusion

The study's findings highlight the need for sustained efforts to improve immunization coverage in Sierra Leone. While there was progress between 2008 and 2013, the subsequent decline emphasizes the importance of maintaining vital immunization programs. Addressing the significant regional disparities, particularly the gap between the southern and northern regions, is recommended. This requires targeted interventions to improve access to immunization services, strengthen health systems, and address the northern region's underlying social and economic factors. Additionally, ongoing monitoring and evaluation are essential to identify and address emerging challenges and ensure equitable access to immunization for all children in Sierra Leone.

## Introduction

The World Health Organization (WHO) highlights childhood immunization as one of the most effective public health interventions to reduce the risk of major infectious diseases in children [1]. Early childhood immunization strengthens immunity against VPDs, reduces mortality, and supports developmental milestones [2,3]. To achieve full immunization, children should receive one dose of Bacillus Calmette-Guerin (BCG) at birth; three doses of polio and pentavalent vaccines at 6, 10, and 14 weeks; and one dose of the measles vaccine at 9 months [4]. The primary determinants of full immunization coverage include demographic factors such as age, education, and marital status, alongside socioeconomic factors like unemployment, poverty, long distances to health centers, and lack of insurance coverage [5–8].

In 2023, the WHO reported that global immunization coverage for fully immunized children stood at 83%, representing the proportion of children who received all recommended vaccine doses within their first year of life [9]. However, coverage varies significantly across continents and countries, particularly in low- and middle-income settings. In sub-Saharan Africa (SSA), results from a study published in 2022 showed that only 44% of children were fully immunized, with coverage ranging from 0% to 78% across countries [10]. These disparities may reflect differences in immunization schedules and health system performance. Inequalities in immunization coverage are often more pronounced in countries experiencing conflict or with a recent history of war, such as Sierra Leone [11].

Over the years, Sierra Leone has made significant progress in improving full immunization coverage, partly due to the establishment of the Expanded Program on Immunization in 1978 [13]. Other interventions include the launch of the Free Healthcare Initiative in 2010, which provides free access to immunization services for children under five, lactating mothers, and pregnant women, as well as mass routine

immunization campaigns [14,15]. Between 2008 and 2019, the proportion of children aged 12–23 months who received all essential vaccines increased from 40% to 56%, while the proportion of children who did not receive any vaccines decreased from 16% to 2% [12]. A 2021 cross-sectional study assessing full immunization coverage among children aged 10–23 months reported that 66% were fully immunized within their first year of life [13]. Despite these improvements, the 2019 Sierra Leone Demographic and Health Survey (SLDHS) revealed significant disparities in immunization coverage, favoring children of mothers with primary education or higher and those residing in Bo District [12]. These disparities highlight the need for this study.

Although previous studies have explored routine immunization coverage and its determinants in Sierra Leone [5,11,13,15–20], none have specifically examined the socioeconomic and geographic inequalities in full immunization coverage among two-year-olds. Understanding these trends and inequalities is crucial for designing targeted public health interventions to improve routine immunization uptake and reduce the burden of VPDs in Sierra Leone. This study aims to examine the trends and investigate the socioeconomic and geographic inequalities in full immunization coverage among two-year-olds in Sierra Leone.

## Method

### Study setting and data source

The study setting was Sierra Leone. Sierra Leone, located on the West Coast of Africa, ranks among the top three countries globally with the worst maternal health outcomes, with infant and under-five mortality rates of 75 per 1,000 live births and 122 per 1,000 live births, respectively [12]. Data from the SLDHS conducted in 2008, 2013, and 2019 were utilised. The SLDHS is a national survey designed to identify trends and variations in social issues, health indicators, and demographics across diverse age groups and genders. The survey utilised a stratified multi-stage cluster sampling method within a cross-sectional framework to select participants. The SLDHS report comprehensively describes the sampling procedure [12]. Our study involved two-year-old children whose mothers provided comprehensive immunization histories during the SLDHS cycles. The SLDHS data from 2008, 2013, and 2019 are available via the WHO Health Equity Assessment Toolkit (HEAT) online platform [20]. The WHO HEAT database was selected due to its specific design for health equity analysis, offering standardized methods and summary measures that align closely with the objectives of this study. HEAT is a software application developed by the World Health Organisation. HEAT aims to facilitate the exploration, analysis, and reporting of data related to health inequalities. This resource utilises data from the Health Inequality Data Repository, enabling users to effectively assess and visualise health disparities. The analysis utilised disaggregated data from the SLDHS indicators available in HEAT. The paper was written following the Strengthening the Reporting of Observational Studies in Epidemiology (STROBE) guidelines [21].

### Variables

The outcome variable was full immunization coverage in two-year-old children. The WHO HEAT classified children who have received all recommended immunizations by age two as completely immunised. In contrast, those who have received fewer, or no vaccines are classified as not completely immunised. Our study used six variables to stratify inequality. The stratifiers were age categories of women (15–19 and 20–49), levels of educational attainment (no education, primary, and secondary and above), economic status (poorest, poorer, middle, richer, and richest), residential location (rural and urban), child sex (female and male), and sub-national regions (East, North, Northwestern, South, and West).

### Data analysis

The online version of the WHO HEAT was employed to analyse health inequality data. The WHO HEAT offers estimates, confidence intervals, and summary metrics of inequality, enabling a comprehensive evaluation of disparities in full immunization

coverage among two-year-old children. Estimates and confidence intervals (CIs) for the prevalence of full immunization coverage in two-year-old children were computed using the specified stratifiers in the HEAT software. Four metrics were utilised to assess inequality: Difference (D), Ratio (R), Population Attributable Fraction (PAF), and Population Attributable Risk (PAR). Difference quantifies the absolute difference in full immunization coverage among two-year-old children across two subgroups by directly comparing their relative rates. Ratio assesses the prevalence of full immunization across two subgroups by calculating the prevalence ratio in one subgroup to that in the other, providing a relative measure of disparity. Both Difference and Ratio are unweighted metrics, indicating that they disregard the population sizes of the subgroups and concentrate exclusively on the two groups under comparison. PAR quantifies the proportion of a health outcome associated with a particular risk factor in the population, whereas PAF represents the percentage of the overall health impact that would be mitigated if the risk factor were absent. These assessments offer insights into the potential effects of diminished inequality on overall health outcomes. Please consult the literature for a detailed discussion on calculating these metrics [22]. Ratio and PAF are comparative metrics used to assess and compare disparities among various factors in relation to each other. Difference and PAR serve as definitive metrics, offering clear values that reflect the precise disparity in full immunization coverage among two-year-old children or the percentage of health outcomes linked to a particular risk factor. This distinction is crucial, as absolute measures such as Difference and PAR provide clear insights into the level of inequality, while relative measures like Ratio and PAF contextualise these disparities within wider population dynamics. The WHO recognised the necessity of providing summary metrics in both absolute and relative formats to formulate policy-relevant conclusions. The summary measurements and calculations of the WHO metrics are comprehensively documented in the literature [23,24].

### Ethics approval and consent to participate

This study did not seek ethical clearance since the WHO HEAT software and the dataset are freely available in the public domain.

## Results

### Trends in full immunization coverage among two-year-olds in Sierra Leone, 2008–2019

Fig 1 illustrates the trends in full immunization coverage among two-year-olds in Sierra Leone from 2008 to 2019. The figure shows an increase in full immunization coverage among two-year-olds in Sierra Leone from 41.1% in 2008 to 73.1% in 2013, indicating successful immunization interventions. However, the sharp decline in full immunization coverage among two-year-olds in Sierra Leone to 50.4% in 2019 suggests the need for continued sustained immunization efforts to improve immunization activities.

Table 1 illustrates the trends in full immunization coverage among two-year-olds in Sierra Leone during the study period disaggregated by the various subgroups of the stratefiers. In 2008, coverage remained notably low, particularly among children from the poorest quintile (33.8%) and those living in the Western province (36.2%). By 2013, coverage was markedly increased across all subgroups, with the poorest quintile achieving 73.6% and children in the Western Province reaching 63.8%, reflecting enhanced efforts in routine immunization programs. The most notable progress was recorded in 2013, with full immunization coverage reaching 73.6% among children in the poorest quintile, 73.2% for those whose mothers had no formal education, and 72.8% for those residing in rural areas. However, by 2019, coverage declined across all dimensions, largely due to the Ebola outbreak during 2014–2016. There was no significant difference in coverage between male and female children. Geographically, all provinces experienced increased coverage except for the northern province, which declined to 36.6%.

### Inequalities in full immunization coverage among two-year-olds in Sierra Leone, 2008–2019

Table 2 presents results of the magnitude of inequality based on the various measures considered in this study (D, R, PAF, PAR). Despite differences in inequality across age of the mother, economic status, educational level, place of

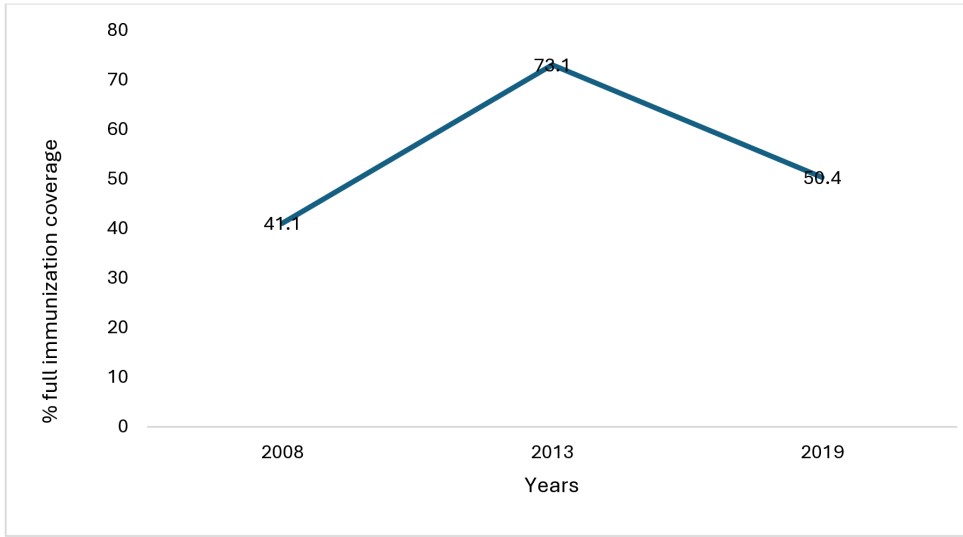

**Fig 1. Trends in full immunization coverage among two-year-olds in Sierra Leone, 2008-2019.**

residence, and sex of the child, between groups and over time, none of the results were significant. At the sub-national level, the coverage difference between children in the southern region and those in the northern region increased from 13.0 percentage points in 2008 to 23.9 percentage points in 2019, with the confidence intervals indicating that this difference was statistically significant. The ratio of children in the southern region compared to those in the northern region also increased from 1.4 percentage points in 2008 to 1.7 percentage points in 2019, with confidence intervals suggesting the presence of inequality between these two groups.

## Discussion

This study explored the trends and socioeconomic and geographical inequalities in full immunization coverage among two-year-olds in Sierra Leone. It revealed an increase in fully immunization coverage from 41.1% in 2008 to 73.1% in 2013, followed by a sharp decline to 50.4% in 2019. Coverage trends improved across various dimensions, including age, economic status, educational levels, place of residence, and sex. However, at the provincial level, all provinces saw increased coverage except for the Northern region, which decreased. Notably, there was better coverage among the poorest populations and those living in rural areas, with inequality primarily existing at the subnational level.

The increase in full immunization coverage among two-year-olds from 2008 to 2013 is encouraging and can be attributed to several government interventions to enhance healthcare services, including routine immunization. Key initiatives included the launch of Free Healthcare Policy (FHP) in 2010 [14] and two nationwide immunization campaigns conducted in 2011 and 2012 [21,22]. This finding aligns with a study on measles immunization coverage inequalities in Sierra Leone, which stated similar reasons for the increase in coverage [23]. The decline in coverage observed in 2019 is likely linked to the Ebola outbreak from 2014 through 2016 [23,24], which disrupted routine immunization services and uptake, consistent with findings from other studies in Sierra Leone [16,25]. Overall, routine immunization coverage rates over the years have remained significantly below the expanded program on immunization target of ≥ 95% [26], indicating that many children remain at increased risk of contracting vaccine-preventable diseases (VPDs). Furthermore, the emergence of the COVID-19 pandemic in Sierra Leone may have further adversely affected routine immunization coverage, increasing the risk of VPDs among children [15]. As Sierra Leone works to rebuild a resilient health system after multiple shocks in recent decades, an outbreak of VPDs would impose an additional burden on healthcare systems, hindering

**Table 1. Trends in full immunization coverage among two-year-olds in Sierra Leone, 2008-2019.**

| Dimension | 2008 | | | | 2013 | | | | 2019 | | | |
|---|---|---|---|---|---|---|---|---|---|---|---|---|
| | N | %. | LB | UB | N | %. | LB | UB | N | %. | LB | UB |
| **Age** | | | | | | | | | | | | |
| 15-19 years | 57 | 39.2 | 25.4 | 55.0 | 129 | 71.2 | 62.2 | 78.8 | 85 | 44.6 | 33.9 | 55.9 |
| 20-49 years | 881 | 41.3 | 37.2 | 45.4 | 1882 | 73.2 | 69.9 | 76.3 | 1581 | 50.7 | 47.5 | 54.0 |
| **Economic status** | | | | | | | | | | | | |
| Quintile 1 (poorest) | 197 | 33.8 | 25.9 | 42.7 | 480 | 73.6 | 68.2 | 78.3 | 378 | 51.2 | 45.5 | 56.9 |
| Quintile 2 | 198 | 42.3 | 33.2 | 51.9 | 470 | 70.7 | 64.5 | 76.2 | 370 | 54.7 | 48.8 | 60.6 |
| Quintile 3 | 233 | 41.9 | 33.7 | 50.5 | 392 | 70.9 | 65.1 | 76.2 | 344 | 53.9 | 47.9 | 59.9 |
| Quintile 4 | 176 | 46.8 | 38.0 | 55.9 | 393 | 75.6 | 69.8 | 80.6 | 309 | 46.9 | 40.9 | 53.0 |
| Quintile 5 (richest) | 135 | 41.5 | 32.8 | 50.8 | 275 | 75.7 | 63.5 | 84.8 | 264 | 42.8 | 34.0 | 52.0 |
| **Education** | | | | | | | | | | | | |
| No education | 732 | 40.0 | 35.1 | 45.1 | 1432 | 73.2 | 69.7 | 76.5 | 889 | 50.5 | 46.6 | 54.4 |
| Primary education | 107 | 50.9 | 39.5 | 62.3 | 276 | 67.6 | 59.4 | 74.8 | 241 | 52.7 | 45.7 | 59.6 |
| Secondary education | 81 | 39.1 | 29.2 | 50.0 | 288 | 76.9 | 70.4 | 82.3 | 495 | 49.4 | 43.8 | 55.0 |
| Higher education | NA | NA | NA | NA | NA | NA | NA | NA | 41 | 47.5 | 30.9 | 64.8 |
| **Place of residence** | | | | | | | | | | | | |
| Rural | 687 | 40.3 | 35.4 | 45.5 | 1543 | 72.8 | 69.3 | 76.1 | 1076 | 52.8 | 49.0 | 56.6 |
| Urban | 252 | 43.3 | 37.2 | 49.7 | 468 | 73.9 | 66.2 | 80.4 | 590 | 46.0 | 40.5 | 51.6 |
| Sex of the child | | | | | | | | | | | | |
| Female | 463 | 43.3 | 37.6 | 49.2 | 1036 | 72.7 | 68.4 | 76.6 | 844 | 50.2 | 46.0 | 54.3 |
| Male | 476 | 39.0 | 33.7 | 44.6 | 975 | 73.5 | 70.1 | 76.7 | 822 | 50.7 | 46.5 | 54.8 |
| **Region** | | | | | | | | | | | | |
| East | 195 | 49.3 | 42.2 | 56.3 | 494 | 76.1 | 70.8 | 80.7 | 356 | 60.5 | 54.1 | 66.7 |
| North | 416 | 37.9 | 31.5 | 44.8 | 745 | 69.9 | 64.6 | 74.7 | 321 | 36.6 | 30.3 | 43.5 |
| Northwestern | NA | NA | NA | NA | NA | NA | NA | NA | 289 | 47.8 | 40.5 | 55.3 |
| South | 200 | 43.0 | 34.2 | 52.3 | 541 | 78.6 | 73.0 | 83.3 | 395 | 60.4 | 54.2 | 66.3 |
| West | 128 | 36.2 | 28.0 | 45.3 | 230 | 63.8 | 50.0 | 75.7 | 305 | 42.6 | 34.6 | 51.0 |

LB: Lower Bound; UB: Upper Bound; N: Sample; NA: Not Available.

efforts to meet the immunization agenda [27] and achieve Sustainable Development Goal 3 by 2030 [28]. To address these challenges, stakeholders are urged to enhance government healthcare spending and ensure adequate resources are allocated for immunization activities. Additionally, it is recommended that community healthcare workers be integrated into immunization efforts and provide incentives to support vaccine delivery, particularly during campaigns in hard-to-reach areas. Lastly, stakeholders should engage community structures, including women's support groups, youth groups, and facility management committees, to promote vaccine uptake initiatives, especially during campaigns.

This study also revealed that children in the poorest quintile had relatively higher full immunization coverage than those in the richest quintile, albeit with the least increase. This contrasts with a multi-country analysis conducted by WHO, which included ten low- and middle-income countries; in that analysis, 9 out of the ten countries showed economic disparities in full immunization coverage for DTP 3 vaccines favoring children in the richest quintile [29]. In Sierra Leone, the higher coverage of full immunization among children from the poorest quintile can largely be attributed to the FHP introduced in 2010. This program, which provides free healthcare services to pregnant women, children under five, and lactating mothers, has significantly boosted the utilization of healthcare services [14]. Another main reason for the observed disparities may be explained by the fact that caregivers of children in the poorest quintile perceive their children to be at a higher

**Table 2. Inequalities in full immunization coverage among two-year-olds in Sierra Leone, 2008-2019.**

| Dimension | Measure | 2008 Estimate (%) | 2008 CI-LB | 2008 CI-UB | 2013 Estimate (%) | 2013 CI-LB | 2013 CI-UB | 2019 Estimate (%) | 2019 CI-LB | 2019 CI-UB |
|---|---|---|---|---|---|---|---|---|---|---|
| **Mother's Age** | D | 2.1 | −13.6 | 17.7 | 2.0 | −6.9 | 10.9 | 6.1 | −5.5 | 17.7 |
| | PAF | 0.3 | 0.3 | 0.3 | 0.2 | 0.2 | 0.2 | 0.6 | 0.6 | 0.6 |
| | PAR | 0.1 | −0.7 | 0.9 | 0.1 | −0.4 | 0.6 | 0.3 | −0.2 | 0.9 |
| | R | 1.1 | 0.7 | 1.6 | 1.0 | 0.9 | 1.2 | 1.1 | 0.9 | 1.5 |
| **Economic Status (Wealth Quintile)** | D | 7.7 | −4.6 | 20.1 | 2.1 | −9.7 | 13.9 | −8.4 | −19.2 | 2.3 |
| | PAF | 0.9 | 0.7 | 1.1 | 3.6 | 3.5 | 3.7 | 0.0 | −0.1 | 0.1 |
| | PAR | 0.4 | −7.3 | 8.1 | 2.6 | −2.1 | 7.4 | 0.0 | −5.5 | 5.5 |
| | R | 1.2 | 0.9 | 1.7 | 1.0 | 0.9 | 1.2 | 0.8 | 0.7 | 1.1 |
| **Education (4 groups)** | D | NA | NA | NA | NA | NA | NA | −2.9 | −21.0 | 15.1 |
| | PAF | NA | NA | NA | NA | NA | NA | 0.0 | −0.3 | 0.3 |
| | PAR | NA | NA | NA | NA | NA | NA | 0.0 | −15.1 | 15.1 |
| | R | NA | NA | NA | NA | NA | NA | 0.9 | 0.6 | 1.4 |
| **Place of Residence** | D | 3.0 | −5.0 | 11.0 | 1.1 | −6.8 | 9.0 | −6.8 | −13.5 | −0.1 |
| | PAF | 5.3 | 5.2 | 5.4 | 1.1 | 1.1 | 1.2 | 0.0 | −0.1 | 0.1 |
| | PAR | 2.2 | −3.0 | 7.4 | 0.8 | −2.7 | 4.3 | 0.0 | −3.2 | 3.2 |
| | R | 1.1 | 0.9 | 1.3 | 1.0 | 0.9 | 1.1 | 0.9 | 0.8 | 1.0 |
| **Sex** | D | −4.3 | −12.2 | 3.7 | 0.9 | −4.4 | 6.1 | 0.5 | −5.4 | 6.3 |
| | PAF | 0.0 | −0.1 | 0.1 | 0.6 | 0.6 | 0.6 | 0.5 | 0.4 | 0.5 |
| | PAR | 0.0 | −3.1 | 3.1 | 0.4 | −1.6 | 2.4 | 0.2 | −2.2 | 2.7 |
| | R | 0.9 | 0.7 | 1.1 | 1.0 | 0.9 | 1.1 | 1.0 | 0.9 | 1.1 |
| **Subnational Region** | D | 13.0 | 1.8 | 24.2 | 14.8 | 0.8 | 28.8 | 23.9 | 14.8 | 33.1 |
| | PAF | 19.8 | 19.6 | 19.9 | 7.6 | 7.5 | 7.6 | 20.1 | 20.0 | 20.2 |
| | PAR | 8.1 | 1.9 | 14.4 | 5.5 | 2.5 | 8.6 | 10.1 | 5.6 | 14.7 |
| | R | 1.4 | 1.0 | 1.8 | 1.2 | 1.0 | 1.5 | 1.7 | 1.3 | 2.0 |

LB: Lower Bound; UB: Upper Bound; D: Difference; NA: Not Available; PAF: Population Attributable Fraction; PAR: Population Attributable Risk; R: Ratio.

risk of VPDs. Due to their limited financial resources to cover medical expenses, they are particularly concerned about their children contracting VPDs, which could worsen their health. This concern may have influenced their health-seeking behaviour regarding vaccine uptake, resulting in improved coverage. To address this disparity, stakeholders are encouraged to enhance ongoing awareness-raising activities through various platforms, including electronic and print media and social media.

Furthermore, children of mothers with no education showed improved coverage compared to those with primary education or higher over time; however, by 2019, the coverage levels were nearly the same. This finding differs from a multilevel analysis involving three Sub-Saharan African countries (Ghana, Kenya, and Côte d'Ivoire) where the authors reported persistent disparities with increased coverage over time for children of educated mothers [30]. In Sierra Leone, the improved coverage among children of uneducated mothers and the minimal difference observed in 2019 may be attributed to regular sensitization activities aimed at women without formal education. For example, the health education units in every district of Sierra Leone carried out routine health education initiatives aimed at pregnant women and nursing mothers during antenatal care sessions. Additionally, the (health education unit) frequently conduct routine awareness campaigns via local community radio stations and community engagement initiatives. These efforts collectively may have contributed to increased awareness of the benefits of immunization uptake, particularly among uneducated women, thereby resulting in higher observed coverage.

Regarding place of residence, children living in rural areas exhibited relatively higher immunization coverage than those in urban areas, except in 2008, when the coverage rates were equal. This finding aligns with studies conducted in The Gambia [31] and a multi-country study involving The Gambia, The Kyrgyz Republic, and Namibia [32]. The authors attributed this difference favoring rural children to the close relationships these children's mothers have with health-care workers, which likely enhances their trust and confidence in vaccine uptake, resulting in higher coverage. Just as observed with economic status, where immunization rates were higher among the poorest quintile, the rise in full immunization coverage in rural areas appears to result from targeted immunization efforts, focusing on marginalized populations. For example, the Ministry of Health in Sierra Leone initiated weekly outreach sessions, during which healthcare workers delivered immunization services with a specific emphasis on underserved communities. This approach likely played a key role in the increased full immunization coverage in rural areas observed in this study. It is therefore recommended that other low- and middle-income countries facing challenges in improving full immunization coverage and reducing inequality gaps adopt similar targeted approaches.

Finally, this study revealed inequalities in full immunization coverage among two-year-olds at the provincial level; the Southern region recorded the highest coverage, while the Northern region decreased over time. The observed regional disparities in full immunization coverage in Sierra Leone can be attributed to several factors. The Southern region, often considered more developed, may have better access to healthcare facilities, a higher density of health workers, and improved infrastructure, all of which contribute to higher immunization rates. In contrast, the Northern region, often characterized by rural areas, may face challenges such as poor road networks, limited healthcare access, and lower health worker density. Additionally, cultural beliefs, misconceptions about vaccines, and socioeconomic factors like poverty and low education levels can influence immunization uptake in specific regions. These factors can collectively hinder immunization efforts in the Northern region, leading to lower coverage rates than in the Southern region. Consequently, this study recommends conducting a mixed-methods or qualitative study to understand the reasons for the regional heterogeneity in full immunization coverage among two-year-olds in Sierra Leone.

## Strengths and limitations

The WHO HEAT online software offers several strengths. Its user-friendly interface allows for the efficient analysis of health data, enabling us to visualize trends and disparities effectively through various graphical representations. The software is specifically designed for health equity assessments, ensuring that the analysis adheres to established methodologies and standards, which enhances the credibility of the findings. However, there are limitations. The software has restricted data input options, potentially limiting the depth of analysis regarding nuanced socioeconomic factors or cultural or local contexts. Additionally, reliance on pre-existing datasets may overlook more recent or localized information, which could impact the relevance of the findings. Furthermore, while HEAT provides valuable insights into health equity, it may not fully capture the complexities of the underlying determinants of immunization coverage, necessitating complementary qualitative research to enrich the understanding of the observed inequalities. Finally, this study did not perform age standardization, which could have been beneficial in accounting for the variations in age distribution across different survey years.

## Conclusion

The study highlights a complex picture of immunization coverage in Sierra Leone. While there was significant progress between 2008 and 2013, the subsequent decline in 2019 underscores the need for sustained and targeted interventions. While inequalities in coverage based on maternal age, place of residence, and child sex were not statistically significant, a substantial disparity emerged between the southern and northern regions. This suggests that future policies and practices should prioritize strengthening immunization services in the northern region, mainly focusing on addressing the underlying factors contributing to the lower coverage. Additionally, continuous monitoring and evaluation of immunization programs

are crucial to identify and address emerging challenges and maintain the progress achieved. Furthermore, future research should consider utilizing raw datasets to implement age standardization, thereby minimizing potential biases within the given population.

## Author contributions

**Conceptualization:** Umaru Sesay, Augustus Osborne, Camilla Bangura, Bright Opoku Ahinkorah.

**Data curation:** Augustus Osborne.

**Formal analysis:** Umaru Sesay, Augustus Osborne.

**Methodology:** Augustus Osborne.

**Supervision:** Bright Opoku Ahinkorah.

**Validation:** Augustus Osborne, Bright Opoku Ahinkorah.

**Writing – original draft:** Umaru Sesay, Augustus Osborne, Hassan Benya, Camilla Bangura, Bright Opoku Ahinkorah.

**Writing – review & editing:** Umaru Sesay, Augustus Osborne, Hassan Benya, Camilla Bangura, Bright Opoku Ahinkorah.

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
