## [Decision Letter · Decision Letter 0]

10 Mar 2025

PONE-D-24-55286Leaving No Child Behind: Addressing Immunization Inequalities among Two-year-olds in Sierra Leone, 2008-2019PLOS ONE

Dear Dr. Osborne,

Thank you for submitting your manuscript to PLOS ONE. After careful consideration, we feel that it has merit but does not fully meet PLOS ONE’s publication criteria as it currently stands. Therefore, we invite you to submit a revised version of the manuscript that addresses the points raised during the review process.

**The study is purely descriptive on one hand using HEAT software, and on the other hand, does not dive into in the context of Sierra Leone to tell a solid story about child vaccination. I urge authors to engage in their knowledge of the ground and challenge current knowledge of child vaccination in the country.**

**Some unobservables could be "culture" which was not included in the analyses. But we know that "cultural" beliefs could undermine the prevalence of child vaccination. **

**Finally, and this is key to this work. Authors compare estimates from 3 DHS datasets. Although the study population is same "0-23 months", the authors didn't do any efforts to standardize before comparisons. The structure of 0-23 months population might be different?**

We look forward to receiving your revised manuscript.

Kind regards,

Zacharie Tsala Dimbuene, Ph.D.

Academic Editor

PLOS ONE

**Journal Requirements:**

1. When submitting your revision, we need you to address these additional requirements. Please ensure that your manuscript meets PLOS ONE's style requirements, including those for file naming. The PLOS ONE style templates can be found at https://journals.plos.org/plosone/s/file?id=wjVg/PLOSOne_formatting_sample_main_body.pdf and https://journals.plos.org/plosone/s/file?id=ba62/PLOSOne_formatting_sample_title_authors_affiliations.pdf 2. Thank you for uploading your study's underlying data set. Unfortunately, the repository you have noted in your Data Availability statement does not qualify as an acceptable data repository according to PLOS's standards. At this time, please upload the minimal data set necessary to replicate your study's findings to a stable, public repository (such as figshare or Dryad) and provide us with the relevant URLs, DOIs, or accession numbers that may be used to access these data. For a list of recommended repositories and additional information on PLOS standards for data deposition, please see https://journals.plos.org/plosone/s/recommended-repositories.

Reviewers' comments:

Reviewer's Responses to Questions

**Comments to the Author**

1. Is the manuscript technically sound, and do the data support the conclusions?

Reviewer #1: Partly

2. Has the statistical analysis been performed appropriately and rigorously? 

Reviewer #1: Yes

3. Have the authors made all data underlying the findings in their manuscript fully available?

Reviewer #1: Yes

4. Is the manuscript presented in an intelligible fashion and written in standard English?

Reviewer #1: No

5. Review Comments to the Author

**Reviewer #1:**  In this study, the authors assess “Leaving No Child Behind: Addressing Immunization Inequalities among Two-Year-Olds in Sierra Leone, 2008-2019." The author investigates an interesting public health problem, particularly for low-income countries. However, the title and the entire document somewhat disagree; from the title “addressing immunization inequality,” I understand that there is an investigation of the intervention provided to minimize the inequality, but what the author did is trends and immunization inequalities. Hence, it is better to amend the title accordingly. Moreover, addressing the following issue can improve the paper quality.

The authors used 'Immunization' and 'Immunisation' alternatively throughout the document; please make it consistent.

Most of the DHS data required appropriate weighting to ensure national representation and was properly adjusted for clustering, strata, and the unequal probability of selection inherent in the survey due to the use of complex survey design in the national demographic health survey. But the authors didn’t consider these points while managing the data.

The outcome variable full immunization is not clearly defined. On line number 159, the authors try to define taking all recommended immunizations by age two. Which is vague; what are the recommended vaccines, and when should these vaccines be administered to the child to be fully immunized? The age of the child at the time of receiving each recommended vaccine will matter to being fully immunized, please provide evidence.

From lines 121 to 122, the authors stated that 'between 2008 and 2019, the coverage of children aged 12-23 months who received all essential vaccines increased from 40% to 56%’. However, in the results section from lines 209 to 2011, 41.1% in 2008 and 50.4% in 2019 were reported using the same year survey data.

On line 233, the authors stated that ‘the most substantial improvements were observed in 2019'; what was your reference? Since you are talking about the trend, your data showed a decline in proportion in all parameters as compared to 2013, so what do you mean by substantial improvement?

I think the authors' justification from lines 311 to 317 for the highest proportion among the poorest quantile in comparison with the richest one is less sound. It is better to see any government policies that favor the poorest population and positively influence vaccination, like policy and social safety nets. In some countries, the government may offer conditional cash transfers or other forms of financial support to encourage vaccination. This type of support can reduce the financial burden on poor families, making it easier for them to access vaccines and other essential health services. This might collinearly work with other multiple disadvantaged groups, like the uneducated, rural, and geographically inaccessible areas.

My final comment is that the authors would benefit from stating why they prefer to use “WHO HEAT software” instead of other strong multi-task analytical software. Here what the authors did was a descriptive analysis with multiple limitations. I hope using other software like R, Stata, etc. will benefit more by visualizing the statistically significant difference between those variables.

6. PLOS authors have the option to publish the peer review history of their article (what does this mean? ). If published, this will include your full peer review and any attached files.

**Do you want your identity to be public for this peer review?** For information about this choice, including consent withdrawal, please see our Privacy Policy .

Reviewer #1: No

---

## [Author Response · Author response to Decision Letter 1]

14 Mar 2025

The Editor

PLOS ONE

12th March 2025

Ref: PONE-D-24-25104

Title: Leaving No Child Behind: Addressing Immunization Inequalities among Two-year-olds in Sierra Leone, 2008-2019

Response to Reviewers' comments

Dear Sir/Madam,

We want to express our sincere thanks for painstakingly reviewing our manuscript and providing valuable comments and suggestions. Please see our point-by-point response to the reviewers' comments and suggestions. Revisions are highlighted with track changes in the revised manuscript.

The study is purely descriptive on one hand using HEAT software, and on the other hand, does not dive into the context of Sierra Leone to tell a solid story about child vaccination. I urge authors to engage in their knowledge of the ground and challenge current knowledge of child vaccination in the country.

Response: We have thoroughly revised the introduction and discussion sections, incorporating additional text to provide greater clarity on the situation in Sierra Leone and to contextualize the study findings more effectively.

Some unobservables could be "culture" which was not included in the analyses. But we know that "cultural" beliefs could undermine the prevalence of child vaccination.

Response: We appreciate the reviewer’s comment on the role of cultural beliefs in influencing child vaccination. While our analysis was limited to data available in the WHO HEAT database, which does not capture cultural factors, we acknowledge their potential impact. We have noted this limitation in the discussion and highlighted the need for future research to explore cultural influences on immunization uptake using qualitative or mixed-method approaches.

Finally, and this is key to this work. Authors compare estimates from 3 DHS datasets. Although the study population is same "0-23 months", the authors didn't do any efforts to standardize before comparisons. The structure of 0-23 months population might be different?

Response: We appreciate this important observation. However, the WHO HEAT database provides only aggregated data, and access to raw data required for age standardization is not available. We have noted this limitation in the manuscript and recommend future studies using raw DHS data to address this concern.

Journal Requirements:

Response: We have formatted our manuscript as per PLOS ONE style requirements.

Response: The dataset used can be accessed at https://dhsprogram.com/data/available-datasets.cfm

Reviewer #1: In this study, the authors assess “Leaving No Child Behind: Addressing Immunization Inequalities among Two-Year-Olds in Sierra Leone, 2008-2019." The author investigates an interesting public health problem, particularly for low-income countries. However, the title and the entire document somewhat disagree; from the title “addressing immunization inequality,” I understand that there is an investigation of the intervention provided to minimize the inequality, but what the author did is trends and immunization inequalities. Hence, it is better to amend the title accordingly. Moreover, addressing the following issue can improve the paper quality.

Response: As recommended, we have revised the title and it read as “Trends and Ineqaulities of full Immunization coverage among Two-year-olds in Sierra Leone, 2008-2019”.

The authors used 'Immunization' and 'Immunisation' alternatively throughout the document; please make it consistent.

Response: We have replaced the text from “immunisation” to “immunization” throughout the text.

Most of the DHS data required appropriate weighting to ensure national representation and was properly adjusted for clustering, strata, and the unequal probability of selection inherent in the survey due to the use of complex survey design in the national demographic health survey. But the authors didn’t consider these points while managing the data.

Response: We acknowledge the importance of applying weights and adjusting for clustering and stratification in analyses of DHS data. However, the WHO HEAT database provides pre-processed and aggregated data for equity analyses, and it does not allow for user-applied weighting or adjustments for complex survey designs. WHO performed these adjustments during the preparation of the datasets.

The outcome variable full immunization is not clearly defined. On line number 159, the authors try to define taking all recommended immunizations by age two. Which is vague; what are the recommended vaccines, and when should these vaccines be administered to the child to be fully immunized? The age of the child at the time of receiving each recommended vaccine will matter to being fully immunized, please provide evidence.

Response: We appreciate your feedback. We would like to clarify that the definition of full immunization was provided in the Introduction section (lines 98–100). To ensure clarity, we have explicitly stated that full immunization is defined as: “Children are considered fully immunized if they have received one dose of Bacillus Calmette-Guerin (BCG) at birth; three doses of polio and pentavalent vaccines at 6, 10, and 14 weeks; and one dose of the measles vaccine at 9 months.”

From lines 121 to 122, the authors stated that 'between 2008 and 2019, the coverage of children aged 12-23 months who received all essential vaccines increased from 40% to 56%’. However, in the results section from lines 209 to 2011, 41.1% in 2008 and 50.4% in 2019 were reported using the same year survey data.

Response: Thank you for pointing out the discrepancy in the coverage figures between the Introduction and Results sections. To clarify, the coverages reported in lines 121–122 of the Introduction specifically refer to children aged 12–23 months, according to the Sierra Leone Demographic and Health Survey (DHS). In contrast, the coverage discussed in the Results section encompasses all children aged 0–24 months.

On line 233, the authors stated that ‘the most substantial improvements were observed in 2019'; what was your reference? Since you are talking about the trend, your data showed a decline in proportion in all parameters as compared to 2013, so what do you mean by substantial improvement?

Response: Thank you for spotting this mismatch. We have revised the sentence appropriately “The most notable progress was recorded in 2013, with full immunization coverage reaching 73.6% among children in the poorest quintile, 73.2% for those whose mothers had no formal education, and 72.8% for those residing in rural areas. However, by 2019, coverage declined across all dimensions, largely due to the Ebola outbreak during 2014–2016”.

I think the authors' justification from lines 311 to 317 for the highest proportion among the poorest quantile in comparison with the richest one is less sound. It is better to see any government policies that favor the poorest population and positively influence vaccination, like policy and social safety nets. In some countries, the government may offer conditional cash transfers or other forms of financial support to encourage vaccination. This type of support can reduce the financial burden on poor families, making it easier for them to access vaccines and other essential health services. This might collinearly work with other multiple disadvantaged groups, like the uneducated, rural, and geographically inaccessible areas.

Response: Thank you for your feedback. As recommended, we have elaborated on potential factors, including the launch of the free healthcare initiative, which may have contributed to the increased rate of full immunization among children from low-income families.

My final comment is that the authors would benefit from stating why they prefer to use “WHO HEAT software” instead of other strong multi-task analytical software. Here what the authors did was a descriptive analysis with multiple limitations. I hope using other software like R, Stata, etc. will benefit more by visualizing the statistically significant difference between those variables.

Response: We chose the WHO HEAT software because it is specifically designed for health equity analysis, providing standardized methods and summary measures aligned with our study objectives. While we acknowledge its limitations for statistical testing, our focus was on exploring trends and inequalities in full immunization coverage. We have clarified this rationale in the "Methods" section and appreciate your suggestion to consider advanced statistical software like R or Stata for future research. The statement in the method read “The WHO HEAT database was selected due to its specific design for health equity analysis, offering standardized methods and summary measures that align closely with the objectives of this study”.

Intext comment from the manuscript

How does this software (WHO HEAT toolkit) integrate the complex survey design of DHS datasets?

Response: Thank you for your thoughtful comment. The WHO HEAT Plus toolkit is purpose-built to manage the intricate survey structure of DHS datasets, accounting for sampling weights, clustering, and stratification. These functionalities are fully embedded in its computations, ensuring that the estimates and inequality measures accurately represent the target population while maintaining statistical reliability.

Is this the right wording for “parameter”

Response: We have changed from “parameters” to dimension”. It reads “The dimensions utilised included age, region, child's gender, socioeconomic status, educational attainment, and residence”

This means what? have you heard of standardization of variables when using multiple similar datasets?

Response: As indicated at the bottom of the table, the NA implies no data was available for that particular year. Regarding whether we have had about standardization, Yes, we have had about it. Unfortunately, the WHO HEAT database provides only aggregated data, and access to raw data required for age standardization is not available making it impossible for us to do it.

We hope that we have adequately addressed the reviewers' comments, and we look forward to receiving a favorable outcome on our paper.

Yours Sincerely,

Augustus Osborne

Corresponding Author

---

## [Editor Report · Decision Letter 1]

18 Jul 2025

Trends and Inequalities of full Immunization coverage among Two-year-olds in Sierra Leone, 2008-2019

PONE-D-24-55286R1

Dear Dr. Osborne,

We’re pleased to inform you that your manuscript has been judged scientifically suitable for publication and will be formally accepted for publication once it meets all outstanding technical requirements.

Kind regards,

Michael Korvink, MA

Guest Editor

PLOS ONE

Additional Editor Comments (optional):

Thank you for your thorough and thoughtful revision of the manuscript titled "Trends and Inequalities of Full Immunization Coverage Among Two-Year-Olds in Sierra Leone, 2008–2019." After reviewing the updated version and your responses to the reviewer comments, I am pleased to inform you that your manuscript has been accepted for publication in PLOS ONE.

The revisions have satisfactorily addressed the concerns raised during the previous review cycle. We appreciate your responsiveness and the effort you invested in improving the submission.
---

## [Editor Report · Acceptance letter]

PONE-D-24-55286R1

PLOS ONE

Dear Dr. Osborne,

I'm pleased to inform you that your manuscript has been deemed suitable for publication in PLOS ONE. Congratulations! Your manuscript is now being handed over to our production team.

Kind regards,

on behalf of

Dr. Michael Korvink

Guest Editor

PLOS ONE